# Gradient Boosted Normalizing Flows

**Robert Giaquinto**          **Arindam Banerjee**
Department of Computer Science & Engineering
University of Minnesota, Twin Cities
Minneapolis, MN 55455, USA

## Abstract

By chaining a sequence of differentiable invertible transformations, normalizing flows (NF) provide an expressive method of posterior approximation, exact density evaluation, and sampling. The trend in normalizing flow literature has been to devise deeper, more complex transformations to achieve greater flexibility. We propose an alternative: Gradient Boosted Normalizing Flows (GBNF) model a density by successively adding new NF components with gradient boosting. Under the boosting framework, each new NF component optimizes a weighted likelihood objective, resulting in new components that are fit to the suitable residuals of the previously trained components. The GBNF formulation results in a mixture model structure, whose flexibility increases as more components are added. Moreover, GBNFs offer a wider, as opposed to strictly deeper, approach that improves existing NFs at the cost of additional training—not more complex transformations. We demonstrate the effectiveness of this technique for density estimation and, by coupling GBNF with a variational autoencoder, generative modeling of images. Our results show that GBNFs outperform their non-boosted analog, and, in some cases, produce better results with smaller, simpler flows.

## 1   Introduction

Deep generative models seek rich latent representations of data, and provide a mechanism for sampling new data. A popular approach to generative modeling is with variational autoencoders (VAEs) [47]. A major challenge in VAEs, however, is that they assume a factorial posterior, which is widely known to limit their flexibility [9, 14, 41, 48, 57, 65, 76, 78]. Further, VAEs do not offer exact density estimation, which is a requirement in many settings.

Normalizing flows (NF) are an important recent development and can be used in both density estimation [19, 67, 73] and variational inference [65]. Normalizing flows are smooth, invertible transformations with tractable Jacobians, which can map a complex data distribution to simple distribution, such as a standard normal [61]. In the context of variational inference, a normalizing flow transforms a simple, known base distribution into a more faithful representation of the true posterior. As such, NFs complement VAEs, providing a method to overcome the limitations of a factorial posterior. Flow-based models are also an attractive approach for density estimation [18, 19, 20, 33, 38, 40, 46, 60, 61, 69, 73] because they provide exact density computation and sampling with only a single neural network pass (in some instances) [24].

Recent developments in NFs have focused of creating deeper, more complex transformations in order to increase the flexibility of the learned distribution [3, 11, 13, 38, 40, 46, 53]. With greater model complexity comes a greater risk of overfitting while slowing down training, prediction, and sampling. Boosting [27, 28, 29, 30, 56] is flexible, robust to overfitting, and generally one the most effective learning algorithms in machine learning [36]. While boosting is typically associated with regression and classification, it is also applicable in the unsupervised setting [8, 8, 34, 35, 52, 57, 68].

**Our contributions.** In this work we propose a *wider*, as opposed to strictly deeper, approach for increasing the expressiveness of density estimators and posterior approximations. Our approach, *gradient boosted normalizing flows* (GBNF), iteratively adds new NF components to a model based on gradient boosting, where each new NF component is fit to the residuals of the previously trained components. A weight is learned for each component of the GBNF model, resulting in a mixture structure. However, unlike a mixture model, GBNF offers the optimality advantages associated with boosting [2], and a simplified training objective that focuses solely on optimizing a single new component at each step. GBNF compliments existing flow-based models, improving performance at the cost of additional training cycles—not more complex transformations. Prediction and sampling are not slowed with GBNF, as each component is independent and operates in parallel.

While gradient boosting is straight-forward to apply in the density estimation setting, our analysis highlights the need for *analytically* invertible flows in order to efficiently boost flow-based models for variational inference. Further, we address the "decoder shock" phenomenon—a challenge unique to VAEs with GBNF approximate posteriors, where the loss increases suddenly coinciding with the introduction of a new component. Our experiments show that augmenting the VAE with a GBNF variational posterior produces image modeling results on par with state-of-the-art NFs. Lastly, GBNF improves density estimation performance on complex, multi-modal data.

## 2  Background

**Normalizing Flows** Tabak and Turner [73], Tabak and Vanden-Eijnden [74] introduce normalizing flows (NF) as a composition of simple maps. Parameterizing flows with deep neural networks [19, 20, 67] has popularized the technique for density estimation and variational inference [61].

**Variational Inference** Rezende and Mohamed [65] use NFs to modify the VAE's [47] posterior approximation $q_0$ by applying a chain of $K$ transformations $\mathbf{z}_K = f_K \circ \cdots \circ f_1(\mathbf{z}_0)$ to the inference network output $\mathbf{z}_0 \sim q_0(\mathbf{z}_0 \mid \mathbf{x})$. By defining $f_k, k = 1, \ldots, K$ as an invertible, smooth mapping, then with the chain rule and inverse function theorem $\mathbf{z}_k = f_k(\mathbf{z}_{k-1})$ has a computable density [73, 74]: $q_k(\mathbf{z}_k) = q_{k-1}(\mathbf{z}_{k-1}) \left| \det \frac{\partial f_k^{-1}}{\partial \mathbf{z}_{k-1}} \right| = q_{k-1}(\mathbf{z}_{k-1}) \left| \det \frac{\partial f_k}{\partial \mathbf{z}_{k-1}} \right|^{-1}$. The VAE maximizes a lower bound on the log-likelihood of the data: the evidence lower bound (ELBO) [5, 44, 79]. Thus, a VAE with a $K$-step flow-based posterior minimizes the negative-ELBO:

$$
\begin{aligned}
\mathcal{F}_{\theta,\phi}^{(VI)}(\mathbf{x}) &= \mathbb{E}_{q_K} \left[ -\log p_\theta(\mathbf{x}, \mathbf{z}_K) + \log q_K(\mathbf{z}_K \mid \mathbf{x}) \right] \\
&= \mathbb{E}_{q_0} \left[ -\log p_\theta(\mathbf{x} \mid \mathbf{z}_K) - \sum_{k=1}^{K} \log \left| \det \frac{\partial f_k}{\partial \mathbf{z}_{k-1}} \right| \right] + KL\left( q_0(\mathbf{z}_0 \mid \mathbf{x}) \,\|\, p(\mathbf{z}_K) \right) , \quad (1)
\end{aligned}
$$

where $q_0(\mathbf{z}_0 \mid \mathbf{x})$ is a known base distribution (e.g. standard normal) with parameters $\phi$.

**Density Estimation** Given a set of samples $\{\mathbf{x}_i\}_{i=1}^{n}$ from a target distribution $p^*$, our goal is to learn a flow-based model $p_\phi(\mathbf{x})$, which corresponds to minimizing the forward KL-divergence: $\mathcal{F}^{(ML)}(\phi) = KL(p^*(\mathbf{x}) \,\|\, p_\phi(\mathbf{x}))$ [61]. A NF formulates $p_\phi(\mathbf{x})$ as a transformation $\mathbf{x} = f(\mathbf{z})$ of a base density $p_0(\mathbf{z})$ with $f = f_K \circ \cdots \circ f_1$ as a $K$-step flow [19, 20, 60]. Thus, to estimate the expectation over $p^*$ we take a Monte Carlo approximation of the forward KL, yielding:

$$
\mathcal{F}^{(ML)}(\phi) \approx -\frac{1}{n} \sum_{i=1}^{n} \left[ \log p_0 \left( f^{-1}(\mathbf{x}_i) \right) + \sum_{k=1}^{K} \log \left| \det \frac{\partial f_k^{-1}}{\partial \mathbf{x}_i} \right| \right] , \quad (2)
$$

which is equivalent to fitting the model to samples $\{\mathbf{x}_i\}_{i=1}^{n}$ by maximum likelihood estimation [61].

**Gradient Boosting** With gradient boosting [28, 29, 30, 56] we consider a loss $\mathcal{F}(G)$, where $G(\cdot)$ is a function representing the current model. To minimize the loss $\mathcal{F}(G)$, we introduce a new component $g$ that is fit to the *functional gradient* $\nabla \mathcal{F}(G)$ at the current model. Choosing the best function $g$ in a class of functions $\mathcal{G}$ (e.g. regression trees), corresponds to solving a linear program where $\nabla \mathcal{F}(G)$ defines the weights for every function in $\mathcal{G}$. Underlying gradient boosting is a connection to conditional gradient descent and the Frank-Wolfe algorithm [25]: we first solve a constrained convex minimization problem to choose $g$, then solve a line-search problem to appropriately weight $g$ relative to the previous components $G$ [8, 35].

| Target | 1 Component | 2 Components | Fine-Tune |

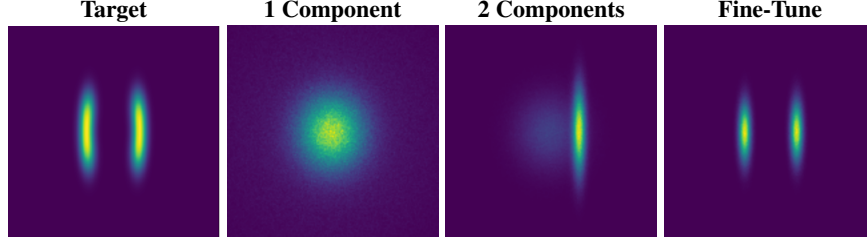

Figure 1: Example of GBNF: A simple affine flow (one scale and shift operation) cannot model the data distribution and leads to mode-covering (**1 Component**). In **2 Components**, GBNF introduces a second component, which seeks a region of high probability that is not well modeled by the first component. Here, fine-tuning components with additional *boosted* training leads to a better solution, shifting the first component to the left ellipsoid and re-weighing appropriately as shown in **Fine-Tune**.

## 3  Gradient Boosted Normalizing Flows for Density Estimation

*Gradient boosted normalizing flows* (GBNF) build on recent ideas in boosting for variational inference [35, 57] and generative models [34] in order to increase the flexibility of density estimators and posteriors approximated with NFs. A GBNF is constructed by successively adding new components based on gradient boosting, where each new component $g_K^{(c)}$ is a $K$-step normalizing flow that is fit to the functional gradient of the loss from the $(c-1)$ previously trained components $G_K^{(c-1)}$.

Gradient boosting assigns a weight $\rho_c$ to the new component $g_K^{(c)}$ and we restrict $\rho_c \in [0, 1]$ to ensure the model stays a valid probability distribution. The resulting density resembles a mixture model:

$$G_K^{(c)}(\mathbf{x}) = \psi\left((1 - \rho_c)\psi^{-1}(G_K^{(c-1)}(\mathbf{x})) + \rho_c\psi^{-1}(g_K^{(c)}(\mathbf{x}))\right)/\Gamma_{(c)} , \qquad (3)$$

where the full model $G_K^{(c)}(\mathbf{x})$ is a monotonic function $\psi$ of a convex combination of fixed components $G_K^{(c-1)}$ and new component $g_K^{(c)}$, and $\Gamma_{(c)}$ is the partition function. Two special cases are of interest: first, when $\psi(a) = a$, which corresponds to an additive mixture model where $\Gamma_{(c)} = 1$. Second, when $\psi(a) = \exp(a)$ with $\psi^{-1}(a) = \log(a)$, corresponding to a multiplicative mixture model [16, 34].

The advantage of GBNF over a standard mixture model with a pre-determined and fixed number of components is that additional components can always be added to the model, and the weights $\rho_c$ for non-informative components degrade to zero. Since each component is a NF, we evaluate (3) recursively using the change of variables formula to expand each component $g_K^{(c)} = p_0\left(f_c^{-1}(\mathbf{x})\right)\prod_{k=1}^{K}|\det\frac{\partial f_{k,c}^{-1}}{\partial \mathbf{x}}|$, where $f_c = f_{c,K} \circ \cdots \circ f_{c,1}$ is the $K$-step flow transformation for component $c$, and the base density $p_0$ is shared by all components.

**Density Estimation**  With GBNF density estimation is similar to (2): we seek flow parameters $\boldsymbol{\phi} = [\phi_1, \ldots, \phi_c]$ that minimize $KL\left(p^*(\mathbf{x}) \| G_K^{(c)}(\mathbf{x})\right)$, which for finite number of samples $\{\mathbf{x}_i\}$ drawn from $p^*(\mathbf{x})$ corresponds to minimizing:

$$\mathcal{F}^{(ML)}(\boldsymbol{\phi}) = -\frac{1}{n}\sum_{i=1}^{n}\left[\log\left\{\psi\left((1 - \rho_c)\psi^{-1}(G_K^{(c-1)}(\mathbf{x}_i)) + \rho_c\psi^{-1}(g_K^{(c)}(\mathbf{x}_i))\right)/\Gamma_{(c)}\right\}\right] . \qquad (4)$$

Directly optimizing (4) for mixture model $G_K^{(c)}$ is non-trivial. Gradient boosting, however, provides an elegant solution that greatly simplifies the problem. During training, the first component is fit using a traditional objective function—no boosting is applied[1]. At stages $c > 1$, we already have $G_K^{(c-1)}$, consisting of a convex combination of the $(c-1)$ $K$-step flow models from the previous stages, and we train a new component $g_K^{(c)}$ by taking a Frank-Wolfe [4, 8, 25, 35, 42, 52] linear approximation of the loss (4). Since jointly optimizing w.r.t. both $g_K^{(c)}$ and $\rho_c$ is a challenging non-convex problem [35], we train $g_K^{(c)}$ until convergence, and then use (4) as the objective to optimize w.r.t the weight $\rho_c$.

## 3.1 Updates to New Boosting Components

**Additive Boosting** We first consider the special case $\psi(a) = a$ and $\Gamma_{(c)} = 1$, corresponding to the additive mixture model. Our goal is to derive an update to the new component $g_K^{(c)}$ using functional gradient descent. Thus, we take the gradient of (4) w.r.t. fixed parameters $\boldsymbol{\phi}_{1:c-1}$ of $G_K^{(c)}$ at $\rho_c \to 0$:

$$\nabla_{\boldsymbol{\phi}_{1:c-1}} \mathcal{F}^{(ML)}(\boldsymbol{\phi})\Big|_{\rho_c \to 0} = -\frac{1-\rho_c}{(1-\rho_c)G_K^{(c-1)} + \rho_c g_K^{(c)}}\Bigg|_{\rho_c \to 0} = -\frac{1}{G_K^{(c-1)}} \tag{5}$$

Maximizing the objective $-\mathcal{F}^{(ML)}(\boldsymbol{\phi})$ is achieved by choosing a new component $g_K^{(c)}$ which is weighted by the negative of the gradient from (5) over the samples. But, since $G_K^{(c-1)}$ is fixed, then the optimization of $g_K^{(c)}$ is a linear program in which $G_K^{(c-1)}(\mathbf{x}_i)$ is a constant—and hence yields a degenerate point probability distribution where the entire probability mass is placed at the minimum point of $G_K^{(c-1)}$. To avoided the degenerate solution, a standard approach [8, 35] adds an entropy regularization term. Thus, optimization of the new component $g_K^{(c)}$ is:

$$g_K^{(c)} = \arg\max_{g_K \in \mathcal{G}_K} \frac{1}{n} \sum_{i=1}^{n} \frac{g_K(\mathbf{x}_i)}{G_K^{(c-1)}(\mathbf{x}_i)} - \lambda \sum_{i=1}^{n} g_K(\mathbf{x}_i) \log g_K(\mathbf{x}_i) , \tag{6}$$

where $\mathcal{G}_K$ is the family of $K$-step flows, and hyperparameter $\lambda > 0$ controls entropy regularization.

**Multiplicative Boosting** In this paper, we instead use $\psi(a) = \exp(a)$ with $\psi^{-1}(a) = \log(a)$, which corresponds to the multiplicative mixture model, and, from the boosting perspective, a multiplicative boosting model [16, 34]. However, in contrast to the existing literature on multiplicative boosting for probabilistic models, we consider boosting with normalizing flow components. In the multiplicative setting, explicitly maintaining the convex combination between $G_K^{(c-1)}$ and $g_K^{(c)}$ is unnecessary: the partition function $\Gamma_{(c)}$ ensures the validity of the probabilistic model. Thus, the multiplicative GBNF seeks a new component $g_K^{(c)}$ and step size $\rho_c$ that minimize:

$$\mathcal{F}^{(ML)}(\boldsymbol{\phi}) = -\frac{1}{n} \sum_{i=1}^{n} \left[ \left( \log(G_K^{(c-1)}(\mathbf{x}_i)) + \rho_c \log(g_K^{(c)}(\mathbf{x}_i)) \right) - \log \Gamma_{(c)} \right] . \tag{7}$$

The objective in (7) represents the loss under the model $G_K^{(c)}$ and derives from minimization of the forward KL-divergence $KL(p^* \| \mathbf{G}_K^{(c)})$, where $\mathbf{G}_K^{(c)}$ is the normalized approximate distribution and $p^*$ the target distribution. To improve (7) with gradient boosting, we show in Appendix B that the difference in losses after introducing a new component $g_K^{(c)}$ to the model is:

$$KL(p^* \| \mathbf{G}_K^{(c-1)}) - KL(p^* \| \mathbf{G}_K^{(c)}) \geq \rho_c \left\{ \mathbb{E}_{p^*}\left[ \log g_K^{(c)}(\mathbf{x}) \right] - \log \mathbb{E}_{G_K^{(c-1)}}\left[ g_K^{(c)}(\mathbf{x}) \right] \right\} , \tag{8}$$

Since $\rho_c \geq 0$, it suffices to focus on the following maximization problem:

$$g_K^{(c)} = \arg\max_{g_K \in \mathcal{G}_K} \mathbb{E}_{p^*}\left[ \log g_K(\mathbf{x}) \right] - \log \mathbb{E}_{G_K^{(c-1)}}\left[ g_K(\mathbf{x}) \right] , \tag{9}$$

for which a direct calculation shows that the solution is given by: $g_K^{(c)}(\mathbf{x}) = \frac{p^*(\mathbf{x})}{G_K^{(c-1)}(\mathbf{x})}$. As we show in Appendix B, our choice of $g_K^{(c)}$ gives a lower bound to (8) with $KL(p^* \| \mathbf{G}_K^{(c)}) \to 0$. The solution to (9) can also be understood in terms of the change-of-measure inequality [1, 22], which forms the basis of the PAC-Bayes bound and a certain regret bounds [1].

Similar to the additive case, the new component chosen in (9) shows that $g_K^{(c)}$ maximizes the likelihood of the samples while discounting those that are already explained by $G_K^{(c-1)}$. Unlike the additive GBNF update in (6), however, the multiplicative GBNF update is a numerically stable and does not require the additional entropy regularization term.

## 3.2 Update to Component Weights

Component weights $\rho$ are updated to satisfy $\rho_c = \arg\min_\rho \mathcal{F}^{(ML)}(\boldsymbol{\phi})$ using line-search. Alternatively, taking the gradient of the loss w.r.t. $\rho_c$ gives a stochastic gradient descent (SGD) algorithm (see Appendix C). Updating $\rho_c$ is only needed once after each component converges.

# 4 Variational Inference with GBNF

Gradient boosting is also applicable to posterior approximation with flow-based models. For variational inference we map a simple base distribution to a complex posterior. Unlike (1), however, we consider a VAE whose approximate posterior $G_K^{(c)}$ is a GBNF with $c$ components and of the form:

$$G_K^{(c)}(\mathbf{z}_K \mid \mathbf{x}) = (1 - \rho_c)G_K^{(c-1)}(\mathbf{z}_K \mid \mathbf{x}) + \rho_c g_K^{(c)}(\mathbf{z}_K \mid \mathbf{x}) . \tag{10}$$

We seek a variational posterior that closely matches the true posterior $p(\mathbf{z}_K \mid \mathbf{x})$, which corresponds to the reverse KL-divergence $KL(G_K^{(c)}(\mathbf{z}_K \mid \mathbf{x}) \,||\, p(\mathbf{z}_K \mid \mathbf{x}))$. Minimizing KL is equivalent to minimizing the negative-ELBO $\mathcal{F}_{\phi,\theta}^{(VI)}(\mathbf{x})$ up to a constant. Thus, we seek to minimize the variational bound:

$$\mathcal{F}_{\phi,\theta}^{(VI)}(\mathbf{x}) = \mathbb{E}_{G_K^{(c)}} \left[ \log G_K^{(c)}(\mathbf{z}_K \mid \mathbf{x}) - \log p_\theta(\mathbf{x}, \mathbf{z}_K) \right] . \tag{11}$$

## 4.1 Updates to New Boosting Components

Given the bound (11), we then derive updates for new components. Similar to Section 3.1, consider the functional gradient w.r.t. $G_K^{(c)}$ at $\rho_c \to 0$:

$$\nabla_{G_K^{(c)}} \mathcal{F}_{\phi,\theta}^{(VI)}(\mathbf{x})\big|_{\rho_c \to 0} = -\log \frac{p_\theta(\mathbf{x}, \mathbf{z}_K)}{G_K^{(c-1)}(\mathbf{z}_K \mid \mathbf{x})} + 1 \tag{12}$$

We minimize $\mathcal{F}_{\theta,\phi}^{(VI)}(\mathbf{x})$ by choosing a new component $g_K^{(c)}$ that has the minimum inner product with the gradient from (12). However, to avoid $g_K^{(c)}$ degenerating to a point mass at the functional gradient's minimum, we add an entropy regularization term[2] controlled by $\lambda > 0$, thus:

$$g_K^{(c)} = \arg\min_{g_K \in \mathcal{G}_K} \sum_{i=1}^{n} \mathbb{E}_{g_K(\mathbf{z}_K|\mathbf{x}_i)} \left[ \nabla_G \mathcal{F}(\mathbf{x}_i) + \lambda \log g_K(\mathbf{z}_K \mid \mathbf{x}_i) \right]. \tag{13}$$

Despite the differences in derivation, optimization of GBNF has a similar structure to other flow-based VAEs. Specifically, with the addition of the entropy regularization, rearranging (13) shows:

$$g_K^{(c)} = \arg\min_{g_K \in \mathcal{G}_K} \mathbb{E}_{g_K(\mathbf{z}|\mathbf{x})} \left[ -\log \frac{p_\theta(\mathbf{x} \mid \mathbf{z}_K^{(c)})}{G_K^{(c-1)}(\mathbf{z}_K^{(c)} \mid \mathbf{x})} \right] + \lambda \cdot KL \left( g_K(\mathbf{z}_K^{(c)} \mid \mathbf{x}) \,||\, p(\mathbf{z}_K^{(c)}) \right) . \tag{14}$$

If $G_K^{(c-1)}$ is constant, then we recover the VAE objective exactly. By learning a GBNF approximate posterior the reconstruction error $-\log p_\theta(\mathbf{x} \mid \mathbf{z}_K^{(c)})$ is down-weighted for samples that are easily explained by the fixed components. For updates to the component weights $\rho$ see Appendix C.

Lastly, we note that during a forward pass the model encodes data to produce $\mathbf{z}_0$. To sample from the posterior $\mathbf{z}_K \sim G_K^{(c)}$, however, we transform $\mathbf{z}_0$ according to $\mathbf{z}_K = f_K^{(j)} \circ \cdots \circ f_1^{(j)}(\mathbf{z}_0)$, where $j \sim Categorical(\rho)$ randomly chooses a component—similar to sampling from a mixture model. Thus, during training we compute a fast stochastic approximation of the likelihood $G_K^{(c)}$. Likewise, prediction and sampling are as fast as the non-boosted setting, and easily parallelizable across components.

## 4.2 Decoder Shock: Abrupt Changes to the VAE Approximate Posterior

Sharing the decoder between all GBNF components presents a unique challenge in training a VAE with a GBNF approximate posterior. During training the decoder acclimates to samples from a particular component (e.g. $g^{(old)}$). However, when a new stage begins the decoder begins receiving samples from a new component $g^{(new)}$. At this point the loss jumps (see Figure 2), a phenomenon we refer to as "decoder shock".

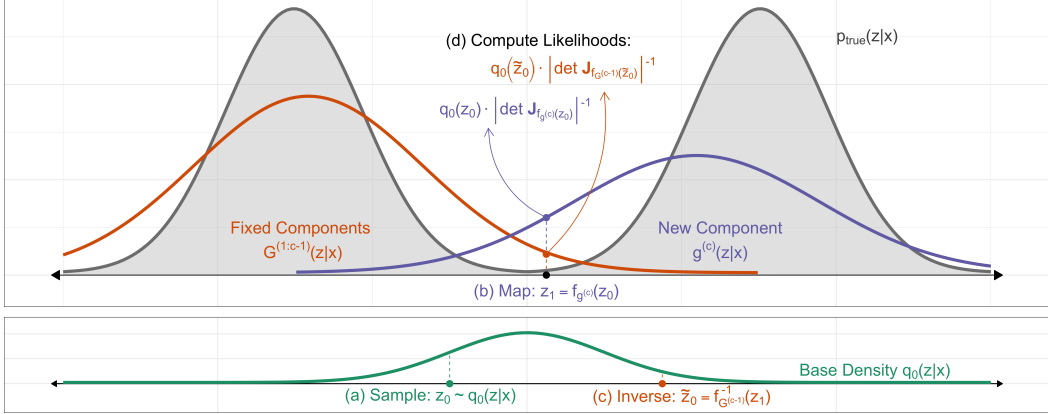

Figure 3: Gradient boosted normalizing flows for variational inference require analytically invertible flows. Similar to a traditional flow-based model: **(a)** samples are drawn from the base density $\mathbf{z}_0 \sim q_0$, and **(b)** transformed by the $K$-step flow transformation. For GBNF, the sample is transformed by the new component giving $\mathbf{z}_1 = f_{g^{(c)}}(\mathbf{z}_0)$. Gradient boosting fits the new component to the *residuals* of the fixed components, and hence requires computing $G^{(c-1)}(\mathbf{z}_1 \mid \mathbf{x})$. Due to the change of variables formula, $G^{(c-1)}(\mathbf{z}_1 \mid \mathbf{x})$ is computed by **(c)** mapping $\mathbf{z}_1$ back to the base density using the inverse flow transformation $\tilde{\mathbf{z}}_0 = f_{G^{(c-1)}}^{-1}(\mathbf{z}_1)$, and then **(d)** evaluating $q_0(\tilde{\mathbf{z}}_0) \cdot |\det \mathbf{J}_{f_{G^{(c-1)}}}|^{-1}$.

The introduction of $g^{(new)}$ causes a sudden shift in the distribution of samples passed to the decoder, causing a sharp increase in reconstruction errors. Further, we anneal the KL [6, 37, 72] in (14) cyclically [31], with restarts corresponding to the introduction of new boosting components, which allows the model to discover useful representations of the data without penalty for complexity. Without KL-annealing, models may ignore $\mathbf{z}$ and rely purely on a powerful decoder [6, 14, 17, 37, 64, 72]. Thus, when the annealing schedule restarts, $g^{(new)}$ is unrestricted and the validation's KL term temporarily increases.

A spike in loss between boosting stages is unique to GBNF. Unlike traditional boosting models, here there is a shared decoder which depends on the boosted components. To overcome the "decoder shock" problem, we periodically sample from the fixed components, helping the

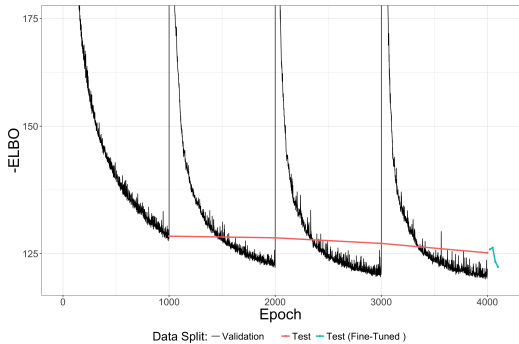

Figure 2: "Decoder Shock" on the Caltech 101 Silhouettes. Test loss (red) decreases steadily by adding new components (every 1000 epochs). However, the loss in batches immediately after adding a new component see a dramatic jump.

decoder *remember* past components. Note, despite drawing samples from $G_K^{(c-1)}$, the parameters for $G_K^{(c-1)}$ remain fixed—samples from $G_K^{(c-1)}$ are purely for the decoder's benefit. Additionally, Figure 2 highlights how *fine-tuning* (blue line) consolidates information from all components and improves results at very little computational cost.

## 5    Experiments

To evaluate GBNF, we highlight results on two toy problems, density estimation on real data, and boosted flows within a VAE for generative modeling of images. We boost coupling flows [20, 46] parameterized by feed-forward networks with TanH activations and a single hidden layer. While RealNVP [20], in particular, is less flexible and shown to be empirically inferior to planar flows in variational inference [65], coupling flows are attractive for boosting: sampling and inference require one forward pass, log-likelihoods are computed exactly, and they are trivially invertible. In

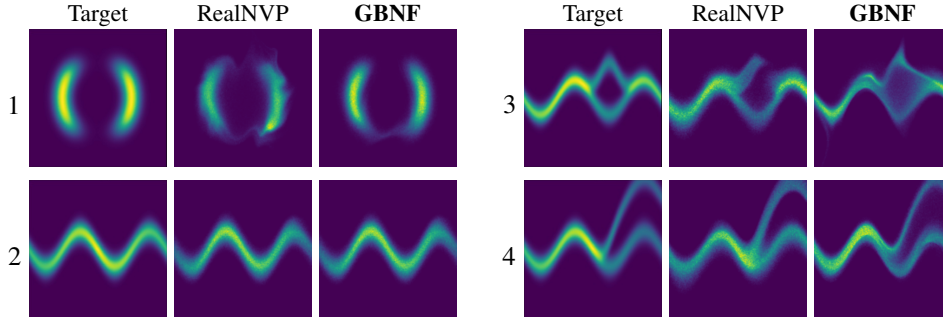

Figure 4: Matching the energy functions from Table 1 of Rezende and Mohamed [65]. The middle columns show deep RealNVPs with $K = 16$ flows. Gradient boosting RealNVP with $c = 2$ components of length $K = 4$ performs as well or better with **half as many parameters**.

the toy experiments flows are trained for 25k iterations using the Adam optimizer [45]. For all other experiments details on the datasets and hyperparameters can be found in Appendix A.

## 5.1 Toy Density Matching

For density matching the model generates samples from a standard normal and transforms them into a complex distribution $p_X$. The 2-dimensional unnormalized target's analytical form $p^*$ is known and parameters are learned by minimizing $KL(p_X \,||\, p^*)$.

**Results** In Figure 4 we compare our results to a deep 16-step RealNVP flow on four energy functions. In each case GBNF provides an accurate density estimation with half as many parameters. When the component flows are flexible enough to model most or all of the target density, components can overlap. However, by training the component weights $\rho$ the model down-weights components that don't provide additional information. On more challenging targets, such as 3 (top-right), GBNF fits one component to each of the top and bottom divergences within the energy function, and some component overlap occurring elsewhere.

## 5.2 Toy Density Estimation

We apply GBNF to the density estimation problems found in [18, 33, 46]. Here the model receives samples from an unknown 2-dimensional data distribution, and the goal is to learn a density estimator of the data. We consider GBNF with either $c = 4$ or 8 RealNVP components, each of which includes $K = 1, 2, 4,$ or 8 coupling layers [20], respectively. Here RealNVP and GBNF use flows of equivalent depth, and we evaluate improvements resulting from GBNF's additional boosted components.

**Results** As shown in Figure 5, even when individual components are weak the composite model is expressive. For example, the 8-Gaussians figure shows that the first component (RealNVP column) fails to model all modes. With additional 1-step flows, GNBF achieves a multimodal density model. Both the 8-Gaussians and Spiral results show that adding boosted components can drastically improve density estimates without requiring more complex transformations. On the Checkerboard and Pinwheel, where RealNVP matches the data more closely, GBNF sharpens density estimates.

## 5.3 Density Estimation on Real Data

Following Grathwohl et al. [33] we report density estimation results on the POWER, GAS, HEP-MASS, and MINIBOONE datasets from the UCI machine learning repository [23], as well as the BSDS300 dataset [55]. We compare boosted and non-boosted RealNVP [20] and Glow models [46]. Glow uses a learned base distribution, whereas our boosted implementation of Glow (and the RealNVPs) use fixed Gaussians. Results for non-boosted models are from [33].

**Results** In Table 1 we find significant improvements by boosting Glow and the more simple RealNVP normalizing flows, even with only $c = 4$ components. Our implementation of Glow was unable to match the results for BSDS300 from [33], and only achieves an average log-likelihood of 152.96

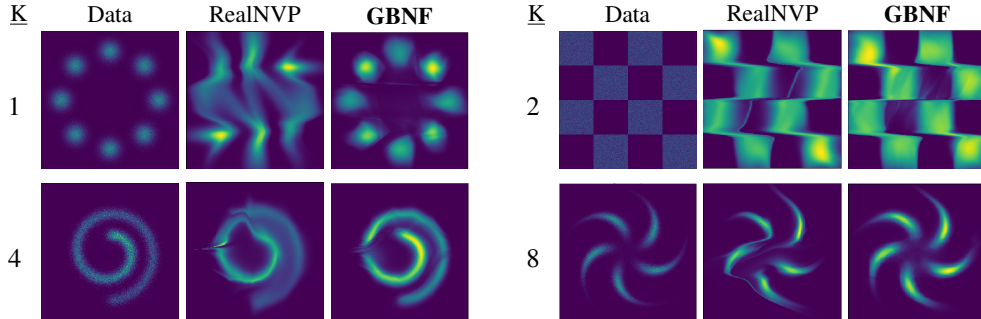

Figure 5: Density estimation for 2D toy data. The **GBNF** columns shows results for a gradient boosted model where each component is a RealNVP flow with $K = 1, 2, 4$ or $8$ flow steps, respectively. For comparison the **RealNVP** column shows results for a single RealNVP model, and is equivalent to GBNF's first component. GBNF models train $c = 4$ components, except on the 8-Gaussians data (top left) where results continued to improve up to 8 components. Results show that GBNF produces more accurate density estimates without increasing the complexity of the flow transformations.

Table 1: Log-likelihood on the test set (higher is better) for 4 datasets from UCI machine learning [23] [55] and BSDS300 [55]. Here $d$ is the dimensionality of data-points and $n$ the size of the dataset. GBNF models include $c = 4$ components. Mean/stdev are estimated over 3 runs.

| Model | POWER↑ | GAS↑ | HEPMASS↑ | MINIBOONE↑ | BSDS300↑ |
|---|---|---|---|---|---|
| | $d=6; n=2,049,280$ | $d=8; n=1,052,065$ | $d=21; n=525,123$ | $d=43; n=36,488$ | $d=63; n=1,300,000$ |
| RealNVP | $0.17_{\pm.01}$ | $8.33_{\pm.14}$ | $-18.71_{\pm.02}$ | $-13.55_{\pm.49}$ | $153.28_{\pm1.78}$ |
| **Boosted RealNVP** | $0.27_{\pm0.01}$ | $9.58_{\pm.04}$ | $-18.60_{\pm0.06}$ | $-10.69_{\pm0.07}$ | $154.23_{\pm2.21}$ |
| Glow | $0.17_{\pm.01}$ | $8.15_{\pm.40}$ | $-18.92_{\pm.08}$ | $-11.35_{\pm.07}$ | $155.07_{\pm.03}$ |
| **Boosted Glow** | $0.24_{\pm0.01}$ | $9.95_{\pm0.11}$ | $-17.81_{\pm0.12}$ | $-10.76_{\pm0.02}$ | $154.68_{\pm0.34}$ |

without boosting. After boosting Glow with $c = 4$ components, however, the log-likelihood rises significantly to 154.68, which is comparable to the baseline.

## 5.4 Image Modeling with Variational Autoencoders

Following Rezende and Mohamed [65], we employ NFs for improving the VAE's approximate posterior [47]. We compare our model on the same image datasets as those used in van den Berg et al. [78]: Freyfaces, Caltech 101 Silhouettes [54], Omniglot [49], and statically binarized MNIST [50].

**Results** In Table 2 we compare the performance of GBNF to other normalizing flow architectures. In all results RealNVP, which is more ideally suited for density estimation tasks, performs the worst of the flow models. Nonetheless, applying gradient boosting to RealNVP improves the results significantly. On Freyfaces, the smallest dataset consisting of just 1965 images, gradient boosted RealNVP gives the best performance—suggesting that GBNF may avoid overfitting. For the larger Omniglot dataset of hand-written characters, Sylvester flows are superior, however, gradient boosting improves the RealNVP baseline considerably and is comparable to Sylvester. GBNF improves on the baseline RealNVP, however both GBNF and IAF's results are notably higher than the non-coupling flows on the Caltech 101 Silhouettes dataset. Lastly, on MNIST we find that boosting improves NLL on RealNVP, and is on par with Sylvester flows. All models have an approximately equal number of parameters, except the baseline VAE (fewer parameters) and Sylvester which has $\approx$ 5x the number of parameters (grid search for hyperparameters is chosen following [78]).

## 6 Related Work

Below we highlight connections between GBNF and related work, along with unique aspects of GBNF. First, we discuss the catalog of normalizing flows that are compatible with gradient boosting. We then compare GBNF to other boosted generative models and flows with mixture formulations.

Table 2: Negative ELBO (lower is better) and Negative log-likelihood (NLL, lower is better) results on MNIST, Freyfaces, Omniglot, and Caltech 101 Silhouettes datasets. For the Freyfaces dataset the results are reported in bits per dim. Results for the other datasets are reported in nats. GBNF models include $c = 4$ RealNVP components. The top 3 NLL results for each dataset are in **bold**.

| Model | MNIST | | Freyfaces | | Omniglot | | Caltech 101 | |
|---|---|---|---|---|---|---|---|---|
| | -ELBO$\downarrow$ | NLL$\downarrow$ | -ELBO$\downarrow$ | NLL$\downarrow$ | -ELBO$\downarrow$ | NLL$\downarrow$ | -ELBO$\downarrow$ | NLL$\downarrow$ |
| VAE | $89.32_{\pm0.07}$ | $84.97_{\pm0.01}$ | $4.84_{\pm0.07}$ | $4.78_{\pm0.07}$ | $109.77_{\pm0.06}$ | $103.16_{\pm0.01}$ | $120.98_{\pm1.07}$ | $108.43_{\pm1.81}$ |
| Planar | $86.47_{\pm0.09}$ | $83.16_{\pm0.07}$ | $4.64_{\pm0.04}$ | $\mathbf{4.60}_{\pm0.04}$ | $105.72_{\pm0.08}$ | $\mathbf{100.18}_{\pm0.01}$ | $116.70_{\pm1.70}$ | $\mathbf{104.23}_{\pm1.60}$ |
| Radial | $88.43_{\pm0.07}$ | $84.32_{\pm0.06}$ | $4.73_{\pm0.08}$ | $4.68_{\pm0.07}$ | $108.74_{\pm0.57}$ | $102.07_{\pm0.50}$ | $118.89_{\pm1.30}$ | $106.88_{\pm1.55}$ |
| Sylvester | $84.54_{\pm0.01}$ | $\mathbf{81.99}_{\pm0.02}$ | $4.54_{\pm0.03}$ | $\mathbf{4.49}_{\pm0.03}$ | $101.99_{\pm0.23}$ | $\mathbf{98.54}_{\pm0.29}$ | $112.26_{\pm2.01}$ | $\mathbf{100.38}_{\pm1.20}$ |
| IAF | $86.46_{\pm0.07}$ | $\mathbf{83.14}_{\pm0.06}$ | $4.73_{\pm0.04}$ | $4.70_{\pm0.05}$ | $106.34_{\pm0.14}$ | $100.97_{\pm0.07}$ | $119.62_{\pm0.84}$ | $108.41_{\pm1.31}$ |
| RealNVP | $88.04_{\pm0.07}$ | $83.36_{\pm0.09}$ | $4.66_{\pm0.17}$ | $4.62_{\pm0.16}$ | $106.22_{\pm0.59}$ | $100.43_{\pm0.19}$ | $123.26_{\pm2.06}$ | $113.00_{\pm1.70}$ |
| GBNF | $87.00_{\pm0.16}$ | $\mathbf{82.59}_{\pm0.03}$ | $4.49_{\pm0.01}$ | $\mathbf{4.41}_{\pm0.01}$ | $105.60_{\pm0.20}$ | $\mathbf{99.09}_{\pm0.17}$ | $121.41_{\pm0.71}$ | $\mathbf{106.40}_{\pm0.54}$ |

**Flows Compatible with Gradient Boosting**    While all normalizing flows can be boosted for density estimation, boosting for variational inference is only practical with *analytically* invertible flows (see Figure 3). While planar and radial [65], Sylvester [78], and neural autoregressive flows [18, 40] are provably invertible, we cannot compute the inverse. Inverse and masked autoregressive flows [48, 60] are invertible, but $D$ times slower to invert where $D$ is the dimensionality of $\mathbf{z}$.

Analytically invertible flows include those based on coupling layers, such as NICE [19], RealNVP [20], and Glow—which replaced RealNVP's permutation operation with a $1 \times 1$ convolution [46]. Neural spline flows increase the flexibility of both coupling and autoregressive transforms using monotonic rational-quadratic splines [24], and non-linear squared flows [80] are highly multi-modal and can be inverted for boosting. Continuous-time flows [10, 12, 33, 69] use transformations described by ordinary differential equations, with FFJORD being "one-pass" invertible by solving an ODE.

**Flows with Mixture Formulations**    The main bottleneck in creating more expressive flows lies in the base distribution and the class of transformation function [61]. Autoregressive [18, 40, 43, 48, 53, 60], residual [3, 13, 65, 78], and coupling-layer flows [19, 20, 38, 46, 63] are the most common classes of finite transformations, however, discrete (RAD, [21]) and continuous (CIF, [15]) mixture formulations offer a promising new approach where the base distribution and transformation change according to the mixture component. GBNF also presents a mixture formulation, but trained in a different way, where only the updates to the newest component are needed during training and extending an existing model with additional components is trivial. Moreover, GBNF optimizes a different objective that fits new components to the residuals of previously trained components, which can refine the *mode covering* behavior of VAEs (see Hu et al. [39]) and maximum likelihood (similar to Dinh et al. [21]). The continuous mixture approach of CIF, however, cannot be used in the variational inference setting to augment the VAE's approximate posterior [15].

**Gradient Boosted Generative Models**    By considering convex combinations of distributions $G$ and $g$, boosting is applicable beyond the traditional supervised setting [8, 16, 34, 35, 51, 52, 68, 75]. In particular, boosting variational inference (BVI, [16, 35, 57]) improves a variational posterior, and boosted generative models (BGM, [34]) constructs a density estimator by iteratively combining sum-product networks. Unlike BVI and BGM our approach addresses the unique algorithmic challenges of boosting applied to flow-based models—such as the need for analytically invertible flows and the "decoder shock" phenomena when enhancing the VAE's approximate posterior with GBNF.

# 7   Conclusion

In this work we introduce *gradient boosted normalizing flows*, a technique for increasing the flexibility of flow-based models through gradient boosting. GBNF, iteratively adds new NF components, where each new component is fit to the residuals of the previously trained components. We show that GBNF can improve results for existing normalizing flows on density estimation and variational inference tasks. In our experiments we demonstrated that GBNF improves over their baseline single component model, without increasing the depth of the model, and produces image modeling results on par with state-of-the-art flows. Further, we showed GBNF models used for density estimation create more flexible distributions at the cost of additional training and not more complex transformations.

## 8  Broader Impact

As a generative model, gradient boosted normalizing flows (GBNF) are suited for a variety of tasks, including the synthesis of new data-points. A primary motivation for choosing GBNF, in particular, is producing a flexible model that can synthesize new data-points quickly. GBNF's individual components can be less complex and thus faster, yet as a whole the model is powerful. Since the components operate in parallel, prediction and sampling can be done quickly—a valuable characteristic for deployment on mobile devices. One limitation of GBNF is the requirement for additional computing resources to train the added components, which can be costly for deep flow-based models. As such, GBNF advantages research laboratories and businesses with access to scalable computing. Those with limited computing resources may find benchmarking or deploying GBNF too costly.

## Acknowledgements

The research was supported by NSF grants OAC-1934634, IIS-1908104, IIS-1563950, IIS-1447566, IIS-1447574, IIS-1422557, CCF-1451986. We thank the University of Minnesota Supercomputing Institute (MSI) for technical support.

## Footnotes

[1]No boosting during the first stage is equivalent to setting $G_K^{(0)}(\mathbf{x})$ to uniform on the domain of $\mathbf{x}$.

[2]In our experiments that augment the VAE with a GBF-based posterior, we find good results setting the regularization $\lambda = 1.0$. In the density estimation experiments, better results are often achieved with $\lambda$ near 0.8.

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
