[Supplementary Material]

# A  Experiment Details

## A.1  Image Modeling

**Datasets**  In Section 5.4, VAEs are modified with GBNF approximate posteriors to model four datasets: Freyfaces[3], Caltech 101 Silhouettes[4] [54], Omniglot[5] [49], and statically binarized MNIST[6] [50]. Details of these datasets are given below.

The Freyfaces dataset contains 1965 gray-scale images of size $28 \times 20$ portraying one man's face in a variety of emotional expressions. Following van den Berg et al. [78], we randomly split the dataset into 1565 training, 200 validation, and 200 test set images.

The Caltech 101 Silhouettes dataset contains 4100 training, 2264 validation, and 2307 test set images. Each image portrays the black and white silhouette of one of 101 objects, and is of size $28 \times 28$. As van den Berg et al. [78] note, there is a large variety of objects relative to the training set size, resulting in a particularly difficult modeling challenge.

The Omniglot dataset contains 23000 training, 1345 validation, and 8070 test set images. Each image portrays one of 1623 hand-written characters from 50 different alphabets, and is of size $28 \times 28$. Images in Omniglot are dynamically binarized.

Finally, the MNIST dataset contains 50000 training, 10000 validation, and 10000 test set images. Each $28 \times 28$ image is a binary, and portrays a hand-written digit.

**Experimental Setup**  We limit the computational complexity of the experiments by reducing the number of convolutional layers in the encoder and decoder of the VAEs from 14 layers to 6. In Table 2 we compare the performance of our GBNF to other normalizing flow architectures. Planar, radial, and Sylvester normalizing flows (SNF) each use $K = 16$, with SNF's bottleneck set to $M = 32$ orthogonal vectors per orthogonal matrix. IAF is trained with $K = 8$ transformations, each of which is a single hidden layer MADE [32] with either $h = 256$ or $512$ hidden units. RealNVP uses $K = 8$ transformations with either $h = 256$ or $h = 512$ hidden units in the Tanh feed-forward network. For all models, the dimensionality of the flow is fixed at $d = 64$.

Each baseline model is trained for 1000 epochs, annealing the KL term in the objective function over the first 250 epochs as in Bowman et al. [6], Sønderby et al. [72]. The gradient boosted models apply the same training schedule to each component. We optimize using the Adam optimizer [45] with a learning rate of $1e - 3$ (decay of 0.5x with a patience of 250 steps). To evaluate the negative log-likelihood (NLL) we use importance sampling (as proposed in Rezende et al. [66]) with 2000 importance samples. To ensure a fair comparison, the reported ELBO for GBNF models is computed by (1)—effectively dropping GBNF's fixed components term and setting the entropy regularization to $\lambda = 1.0$.

**Model Architectures**  In Section 5.4, we compute results on real datasets for the VAE and VAEs with a flow-based approximate posterior. In each model we use convolutional layers, where convolutional layers follow the PyTorch convention [62]. The encoder of these networks contains the following layers:

$$\text{Conv}(\text{in} = 1, \text{out} = 16, \text{k} = 5, \text{p} = 2, \text{s} = 2)$$
$$\text{Conv}(\text{in} = 16, \text{out} = 32, \text{k} = 5, \text{p} = 2, \text{s} = 2)$$
$$\text{Conv}(\text{in} = 32, \text{out} = 256, \text{k} = 7, \text{p} = 0, \text{s} = 1)$$

where k is a kernel size, p is a padding size, and s is a stride size. The final convolutional layer is followed by a fully-connected layer that outputs parameters for the diagonal Gaussian distribution and amortized parameters of the flows (depending on model).

Similarly, the decoder mirrors the encoder using the following transposed convolutions:

$$\text{ConvT}(\text{in} = 64, \text{out} = 32, \text{k} = 7, \text{p} = 0, \text{s} = 2)$$
$$\text{ConvT}(\text{in} = 32, \text{out} = 16, \text{k} = 5, \text{p} = 0, \text{s} = 2)$$
$$\text{ConvT}(\text{in} = 16, \text{out} = 16, \text{k} = 5, \text{p} = 1, \text{s} = 1, \text{op} = 1)$$

where $\text{op}$ is an outer padding. The decoders final layer is passed to standard 2-dimensional convolutional layer to reconstruction the output, whereas the other convolutional layers listed above implement a gated action function:

$$\mathbf{h}_l = (\mathbf{W}_l * \mathbf{h}_{l-1} + \mathbf{b}_l) \odot \sigma(\mathbf{V}_l * \mathbf{h}_{l-1} + \mathbf{c}_l),$$

where $\mathbf{h}_{l-1}$ and $\mathbf{h}_l$ are inputs and outputs of the $l$-th layer, respectively, $\mathbf{W}_l, \mathbf{V}_l$ are weights of the $l$-th layer, $\mathbf{b}_l, \mathbf{c}_l$ denote biases, $*$ is the convolution operator, $\sigma(\cdot)$ is the sigmoid activation function, and $\odot$ is an element-wise product.

### A.2 Density Estimation on Real Data

**Dataset**   For the unconditional density estimation experiments we follow Papamakarios et al. [60], Uria et al. [77], evaluating on four dataset from the UCI machine learning repository [23] and patches of natural images from natural images [55]. From the UCI repository the POWER dataset ($d = 6$, $N =$2,049,280) contains electric power consumption in a household over a period of four years, GAS ($d = 8$, $N =$1,052,065) contains logs of chemical sensors exposed to a mixture of gases, HEPMASS ($d = 21$, $N =$525,123) contains Monte Carlo simulations from high energy physics experiments, MINIBOONE ($d = 43$, $N =$36,488) contains electron neutrino and muon neutrino examples. Lastly we evaluate on BSDS300, a dataset ($d = 63$, $N =$1,300,000) of patches of images from the homonym dataset. Each dataset is preprocessed following Papamakarios et al. [60].

**Experimental Setup**   We compare our results against Glow [46], and RealNVP [20]. We train models using a small grid search on the depth of the flows $K \in \{5, 10\}$, the number of hidden units in the coupling layers $H \in \{10d, 20d, 40d\}$, where $d$ is the input dimension of the data-points. We trained using a cosine learning rate schedule with the learning rate determined using the learning rate range test [71] for each dataset, and similar to Durkan et al. [24] we use batch sizes of 512 and up to 400,000 training steps, stopping training early after 50 epochs without improvement. The log-likelihood calculation for GBNF follows (7), that is we recursively compute and combine log-likelihoods for each component.

## B   Multiplicative Boosting for Density Estimation

The multiplicative GBNF seeks a new component $g_K^{(c)}$ that minimizes:

$$\mathcal{F}^{(ML)}(\boldsymbol{\phi}) = -\frac{1}{n}\sum_{i=1}^{n}\left[\left(\log(G_K^{(c-1)}(\mathbf{x}_i)) + \rho_c\log(g_K^{(c)}(\mathbf{x}_i))\right) - \log\Gamma_{(c)}\right], \qquad (15)$$

where the partition function $\Gamma_{(c)}$ ensures the validity of the probabilistic model, and, here in the multiplicative setting, makes explicitly maintaining the convex combination between $G_K^{(c-1)}$ and $g_K^{(c)}$ unnecessary. Since the partition function ensures proper normalization, the component weight $\rho_c \in [0, 1]$ simply acts as a step size.

### B.1   Partition Function

First, note that the partition function is defined as $\Gamma_{(c)} = \int_x \prod_{j=1}^{c} (g_K^{(j)})^{\rho_j}(\mathbf{x}) p_0(\mathbf{x}) d\mathbf{x}$ and computing $\Gamma_{(c)}$ for GBNF is straightforward since normalizing flows learn self-normalized distributions—and hence can be computed without resorting to simulated annealing or Markov chains [34]. Moreover, following standard properties [16, 34], we also inherit a recursive property of the partition function. To see the recursive property, denote the un-normalized GBNF density as $\tilde{G}_K^{(c)}$, where $\tilde{G}_K^{(c)} \propto G_K^{(c)}$ but $\tilde{G}_K^{(c)}$ does not integrate to 1. Then, by definition:

$$\Gamma_{(c)}G_K^{(c)}(\mathbf{x}) = g_K^{(c)}(\mathbf{x})^{\rho_c}\tilde{G}_K^{(c-1)}(\mathbf{x}) = \Gamma_{(c-1)}g_K^{(c)}(\mathbf{x})^{\rho_c}G_K^{(c-1)}(\mathbf{x}),$$

then, integrating both sides and using $\int_{\mathbf{x}} G_K^{(c)}(\mathbf{x})d\mathbf{x} = 1$ gives

$$\Gamma_{(c)} = \Gamma_{(c-1)} \int_{\mathbf{x}} g_K^{(c)}(\mathbf{x})^{\rho_c} G_K^{(c-1)}(\mathbf{x})d\mathbf{x} = \Gamma_{(c-1)}\mathbb{E}_{G_K^{(c-1)}}\left[g_K^{(c)}(\mathbf{x})^{\rho_c}\right] \ .$$

and therefore $\Gamma_{(c)} = \Gamma_{(c-1)}\mathbb{E}_{G_K^{(c-1)}}[g_K^{(c)}(\mathbf{x})^{\rho_c}]$, as desired.

## B.2 Deriving the New Component Update

The objective in (15) represents the loss under the model $G_K^{(c)}$ which followed from minimizing the forward KL-divergence $KL(p^*\|\mathbf{G}_K^{(c)})$, where $\mathbf{G}_K^{(c)}$ is the normalized approximate distribution and $p^*$ the target distribution. To improve (15) with gradient boosting, consider the difference in losses after introducing a new component $g_K^{(c)}$ to the model:

$$
\begin{aligned}
KL(p^*\|\mathbf{G}_K^{(c-1)}) - KL(p^*\|\mathbf{G}_K^{(c)}) &= \mathbb{E}_{p^*}\left[\log \frac{p^*(\mathbf{x})}{G_K^{(c-1)}(\mathbf{x})} - \log \frac{p^*(\mathbf{x})}{G_K^{(c)}(\mathbf{x})}\right] \\
&= \mathbb{E}_{p^*}\left[\log \frac{G_K^{(c)}(\mathbf{x})}{G_K^{(c-1)}(\mathbf{x})}\right] \\
&= \mathbb{E}_{p^*}\left[\log \frac{(g_K^{(c)}(\mathbf{x}))^{\rho_c}\tilde{G}_K^{(c-1)}(\mathbf{x})}{\Gamma_{c-1}\mathbb{E}_{G_K^{(c-1)}}[(g_K^{(c)}(\mathbf{x}))^{\rho_c}]} \times \frac{\Gamma_{c-1}}{\tilde{G}_K^{(c-1)}(\mathbf{x})}\right] \\
&= \mathbb{E}_{p^*}\left[\log \frac{(g_K^{(c)}(\mathbf{x}))^{\rho_c}}{\mathbb{E}_{G_K^{(c-1)}}[(g_K^{(c)}(\mathbf{x}))^{\rho_c}]}\right] \\
&= \mathbb{E}_{p^*}\left[\log g_K^{(c)}(\mathbf{x}))^{\rho_c}\right] - \log \mathbb{E}_{G_K^{(c-1)}}\left[g_K^{(c)}(\mathbf{x}))^{\rho_c}\right] \\
&\overset{(a)}{\geq} \mathbb{E}_{p^*}\left[\log g_K^{(c)}(\mathbf{x}))^{\rho_c}\right] - \log \left(\mathbb{E}_{G_K^{(c-1)}}[g_K^{(c)}(\mathbf{x})]\right)^{\rho_c} \\
&= \rho_c\left\{\mathbb{E}_{p^*}\left[\log g_K^{(c)}(\mathbf{x})\right] - \log \mathbb{E}_{G_K^{(c-1)}}\left[g_K^{(c)}(\mathbf{x})\right]\right\} \ , \quad (16)
\end{aligned}
$$

where (a) follows by Jensen's inequality since $\rho_c \in [0,1]$. Note that we want to choose the new component $g_K^{(c)}(\mathbf{x})$ so that $KL(p^*\|\mathbf{G}_K^{(c)})$ is minimized, or equivalently, for a fixed $\mathbf{G}_K^{(c-1)}$, the difference $KL(p^*\|\mathbf{G}_K^{(c-1)}) - KL(p^*\|\mathbf{G}_K^{(c)})$ is maximized. Since $\rho_c \geq 0$, it suffices to focus on the following maximization problem:

$$g_K^{(c)} = \underset{g_K \in \mathcal{G}_K}{\arg\max} \ \mathbb{E}_{p^*}\left[\log g_K(\mathbf{x})\right] - \log \mathbb{E}_{G_K^{(c-1)}}\left[g_K(\mathbf{x})\right] \ . \tag{17}$$

If we choose a new component according to:

$$g_K^{(c)}(\mathbf{x}) = \frac{p^*(\mathbf{x})}{G_K^{(c-1)}(\mathbf{x})} \ , \tag{18}$$

then, with this choice of $g_K^{(c)}$, we see that (17) reduces to:

$$\mathbb{E}_{p^*}\left[\log \frac{p^*(\mathbf{x})}{G_K^{(c-1)}(\mathbf{x})}\right] - \log \mathbb{E}_{G_K^{(c-1)}}\left[\frac{p^*(\mathbf{x})}{G_K^{(c-1)}(\mathbf{x})}\right] = KL\left(p^*\|\mathbf{G}_K^{(c-1)}\right) - \underbrace{\log \mathbb{E}_{p^*(\mathbf{x})}[1]}_{0} \ .$$

Our choice of $g_K^{(c)}$, therefore, gives a lower bound to (16) and as $KL(p^*\|\mathbf{G}_K^{(c)}) \to 0$ the optimization in (18) approaches the maximum achievable value. The solution to (17) can also be understood in terms of the change-of-measure inequality [1, 22], which also forms the basis of the PAC-Bayes bound and a certain regret bounds [1].

**Connection to Boosted Generative Models** The solution in (18) not only gives an insightful description of GBNF as fitting new components to a re-weighted data distribution, but also clarifies the derivation of the broader class Boosted Generative Models [34]. More specifically, Grover and Ermon [34] train the new boosting component $g^{(c)}$ to perform maximum likelihood estimation over a re-weighted data distribution:

$$g^{(c)} = \arg\min_{g \in \mathcal{G}} \mathbb{E}_{\mathcal{D}^{(c-1)}} \left[ -\log g \right] \tag{19}$$

where $\mathcal{D}^{(c-1)}$ denotes a re-weighted data distribution whose samples are drawn with replacement using sample weights inversely-proportional to $G_K^{(c-1)}$. In Grover and Ermon [34] the re-weighted data distribution may include a re-weighting coefficient $\beta \in [0, 1]$ such that:

$$\mathcal{D}^{(c-1)}(\mathbf{x}) = \left( \frac{c_0 p^*(\mathbf{x})}{G^{(c-1)}(\mathbf{x})} \right)^{\beta} , \tag{20}$$

where $c_0$ is the proportionality constant associated with the re-weighted data distribution. In our analysis it suffices to leave $\beta = 1$. We show that, when properly bounded, the objective in (19) shares an optimal solution with our results from (17).

First, note that $g \in \mathcal{G}$ needs to be a bounded measurable function, otherwise one can minimize (19) by scaling $g$. Hence, (19) is a constrained optimization problem, whose Lagrangian is given by:

$$\mathcal{L}(g, \lambda) = \int_{\mathbf{x}} (-\log g(\mathbf{x})) \frac{c_0 p^*(\mathbf{x})}{G^{(c-1)}(\mathbf{x})} d\mathbf{x} + \lambda \left( \int_{\mathbf{x}} g(\mathbf{x}) d\mathbf{x} - c_1 \right) , \tag{21}$$

where $c_1$ determines the scaling of $g$, e.g., when $c_1 = 1$, $g$ will be a probability density function. From the Lagrangian, the optimality condition for any specific $g(\mathbf{x})$ yields:

$$-\frac{1}{\hat{g}(\mathbf{x})} \frac{c_0 p^*(\mathbf{x})}{G^{(c-1)}(\mathbf{x})} + \lambda = 0 \quad \Rightarrow \quad \hat{g}(\mathbf{x}) = \frac{c_0}{\lambda} \frac{p^*(\mathbf{x})}{G^{(c-1)}(\mathbf{x})} . \tag{22}$$

Further, the optimality condition for $\lambda$ yields

$$\int_x \hat{g}(\mathbf{x}) d\mathbf{x} = c_1 \quad \Rightarrow \quad \frac{c_0}{\lambda} \int_x \frac{p^*(\mathbf{x})}{G^{(c-1)}(\mathbf{x})} d\mathbf{x} = c_1 \quad \Rightarrow \quad \lambda = \frac{c_0}{c_1} \int_x \frac{p^*(\mathbf{x})}{G^{(c-1)}(\mathbf{x})} d\mathbf{x} , \tag{23}$$

a positive constant. Hence, the optimal solution $\hat{g}(\mathbf{x})$ for (19) is proportional to (18), the optimal solution to (17).

**Surrogate Losses** Further, our analysis reveals the source of a surrogate loss function [2, 58, 59] which optimizes the global objective—namely, when written as as a minimization problem (18) corresponds to the *weighted* negative log-likelihood of the samples. Surrogate loss functions are common in the boosting framework [2, 7, 26, 27, 28, 68, 70, 75]. Adaboost [26, 27], in particular, solves a re-weighted classification problem where weak learners, in the form of decision trees, optimize surrogate losses like information gain or Gini index. The negative log-likelihood is specifically chosen as a surrogate loss function in other boosted probabilistic and density estimation models which also have $f$-divergence based global objectives [34, 68], however here we clarify that the surrogate loss follows from (16).

**Convergence** Lastly, we note that the analysis of Cranko and Nock [16] highlights important properties of the broader class of boosted density estimation models that optimize (4), of which both the additive and multiplicative forms of GBNF are members. Specifically, Remark 3 in Cranko and Nock [16] shows a sharper decrease in the loss—that is, for any step size $\rho \in [0, 1]$ the loss has geometric convergence:

$$KL\left( p^* \| \mathbf{G}_K^{(c)} | \rho_c \right) \leq (1 - \rho_c) KL\left( p^* \| \mathbf{G}_K^{(c-1)} \right) \tag{24}$$

where $\mathbf{G}_K^{(c)} | \rho_c$ denotes the explicit dependence of $\mathbf{G}_K^{(c)}(\mathbf{x})$ on $\rho_c$. Thus GBNF provides a strong convergence guarantee on the global objective.

**Algorithm 1:** Updating Mixture Weight $\rho_c$.

---

Let: Tolerance $\epsilon > 0$, and Step-size $\delta > 0$

Initialize weight $\rho_c^{(0)} = 1/C$

Set iteration $t = 0$

**while** $|\rho_c^{(t)} - \rho_c^{(t-1)}| < \epsilon$ **do**

    Draw mini-batch samples $\mathbf{z}_{K,i}^{(c-1)} \sim G_K^{(c-1)}(\mathbf{z}_K \mid \mathbf{x}_i)$ and $\mathbf{z}_{K,i}^{(c)} \sim g_K^{(c)}(\mathbf{z}_K \mid \mathbf{x}_i)$ for $i = 1, \dots, n$

    Compute Monte Carlo estimate of gradient

    $\nabla_{\rho_c} \mathcal{F}_{\theta,\phi}^{(VI)}(\mathbf{x}) = \frac{1}{n} \sum_{i=1}^{n} \gamma_{\rho_c}^{(t-1)}(\mathbf{z}_{K,i}^{(c)} \mid \mathbf{x}_i) - \gamma_{\rho_c}^{(t-1)}(\mathbf{z}_{K,i}^{(c-1)} \mid \mathbf{x}_i)$

    t = t + 1

    $\rho_c^{(t)} = \rho_c^{(t-1)} - \delta \nabla_{\rho_c}$

    $\rho_c^{(t)} = \text{clip}(\rho_c^{(t)}, [0,1])$

**return** $\rho_c^{(t)}$

---

## C    Updating Component Weights for Variational Inference

After $g_K^{(c)}(\mathbf{z}_K \mid \mathbf{x})$ has been estimated, the mixture model still needs to estimate $\rho_c \in [0,1]$. Similar to the density estimation setting, the weights on each component can be updated by taking the gradient of the loss $\mathcal{F}_{\phi,\theta}^{(VI)}(\mathbf{x})$ with respect to $\rho_c$. Recall that $G_K^{(c)}(\mathbf{z}_K \mid \mathbf{x})$ can be written as the convex combination:

$$
\begin{aligned}
G_K^{(c)}(\mathbf{z}_K \mid \mathbf{x}) &= (1 - \rho_c) G_K^{(c-1)}(\mathbf{z}_K \mid \mathbf{x}) + \rho_c g_K^{(c)}(\mathbf{z}_K \mid \mathbf{x}) \\
&= \rho_c \left( g_K^{(c)}(\mathbf{z}_K \mid \mathbf{x}) - G_K^{(c-1)}(\mathbf{z}_K \mid \mathbf{x}) \right) + G_K^{(c-1)}(\mathbf{z}_K \mid \mathbf{x}) ,
\end{aligned}
$$

Then, with $\Delta_K^{(c)}(\mathbf{z}_K \mid \mathbf{x}) \triangleq g_K^{(c)}(\mathbf{z}_K \mid \mathbf{x}_i) - G_K^{(c-1)}(\mathbf{z}_K \mid \mathbf{x}_i)$, the objective function $\mathcal{F}_{\theta,\phi}^{(VI)}(\mathbf{x})$ can be written as a function of $\rho_c$:

$$
\begin{aligned}
\mathcal{F}_{\theta,\phi}^{(VI)}(\mathbf{x}) = &\sum_{i=1}^{n} \left\langle \rho_c \Delta_K^{(c)}(\mathbf{z}_K \mid \mathbf{x}_i) + G_K^{(c-1)}(\mathbf{z}_K \mid \mathbf{x}_i), -\log p_\theta(\mathbf{x}_i, \mathbf{z}_K) \right\rangle \\
&+ \sum_{i=1}^{n} \left\langle \rho_c \Delta_K^{(c)}(\mathbf{z}_K \mid \mathbf{x}_i) + G_K^{(c-1)}(\mathbf{z}_K \mid \mathbf{x}_i), \log \left( \rho_c \Delta_K^{(c)}(\mathbf{z}_K \mid \mathbf{x}_i) + G_K^{(c-1)}(\mathbf{z}_K \mid \mathbf{x}_i) \right) \right\rangle .
\end{aligned}
\tag{25}
$$

The above expression can be used in a black-box line search method or, as we have done, in a stochastic gradient descent algorithm 1. Toward that end, taking gradient of (25) w.r.t. $\rho_c$ yields the component weight updates:

$$
\frac{\partial \mathcal{F}_{\phi,\theta}^{(VI)}}{\partial \rho_c} = \sum_{i=1}^{n} \left( \underset{g_K^{(c)}(\mathbf{z}_K \mid \mathbf{x}_i)}{\mathbb{E}} \left[ \gamma_{\rho_c}^{(t-1)}(\mathbf{z}_K \mid \mathbf{x}_i) \right] - \underset{G_K^{(c-1)}(\mathbf{z}_K \mid \mathbf{x}_i)}{\mathbb{E}} \left[ \gamma_{\rho_c}^{(t-1)}(\mathbf{z}_K \mid \mathbf{x}_i) \right] \right) ,
\tag{26}
$$

where we've defined:

$$
\gamma_{\rho_c}^{(t-1)}(\mathbf{z}_K \mid \mathbf{x}_i) \triangleq \log \left( \frac{(1 - \rho_c^{(t-1)}) G_K^{(c-1)}(\mathbf{z}_K \mid \mathbf{x}_i) + \rho_c^{(t-1)} g_K^{(c)}(\mathbf{z}_K \mid \mathbf{x}_i)}{p_\theta(\mathbf{x}_i, \mathbf{z}_K)} \right) .
$$

To ensure a stable convergence we follow Guo et al. [35] and implement an SGD algorithm with a decaying learning rate.

Updating a component's weight is only needed once after each component converges. We find, however, that results improve by "fine-tuning" each component and their weights with additional training after the initial training pass. During the fine-tuning stage, we sequentially retrain each component $g_K^{(i)}$ for $i = 1, \dots, c$, during which we treat $G_K^{(-i)}$ as fixed where $-i$ represents the mixture of all other components: $1, \dots, i-1, i+1, \dots c$. Figure 1 demonstrates this phenomenon: when a

single flow is not flexible enough to model the target, mode-covering behavior arises. Introducing the second component trained with the boosting objective improves results, and consequently the second component's weight is increased. Fine-tuning the first component leads to a better solution and assigns equal weight to the two components.

## Footnotes

[3]http://www.cs.nyu.edu/~roweis/data/frey_rawface.mat

[4]https://people.cs.umass.edu/~marlin/data/caltech101_silhouettes_28_split1.mat

[5]https://github.com/yburda/iwae/tree/master/datasets/OMNIGLOT

[6]http://yann.lecun.com/exdb/mnist/