[Reviews · NeurIPS 2020]

Review 1

Summary and Contributions: The paper proposes to use gradient boosting to learn mixtures of normalizing flows. Like for gradient boosting, each new component is fit in a greedy fashion the "residuals" of the previously trained components. The claim is that for increasing expressiveness, using Gradient Boosted Normalizing Flows (GBNFs) to learn mixtures of flows might be a viable alternative to learning deeper flows (as is the common approach).

Strengths: The method proposed provides a simple approach to improve normalizing flow models by iteratively adding new normalizing flow components in a mixture model.

Weaknesses: There appears to be little novelty in the paper. Several papers have considered boosting for density estimation (e.g. [27, 52]) and variational inference (e.g. [6, 28, 41, 46]). It is not clear how this work differs from these. It is stated in the background section that: "Unlike BVI and BGM our approach extends boosting to flow-based models: enhancing the VAE’s approximate posterior, and fitting flexible density estimators." If this is the only difference, this is merely a special case of previous work. It is also stated in the background section that: "[...] boosted generative models (BGM, [27]) constructs a density estimator by iteratively combining sum-product networks." However, consulting the BGM paper, it is stated that: "Our meta-algorithm is general and can construct ensembles of any existing generative model that permits (approximate) likelihood evaluation", while sum-product networks are used in experiments. Moreover, it is stated in the BGM paper that: "The GenBGM algorithm can be adapted to normalizing flow models whereby every transformation is interpreted as a weak learner." In the experiments (Fig 3, Fig 5, Table 1), it is not too surprising that more components improves performance as this increases the expressiveness of the model. It would be more instructive to consider e.g. (C,K)=(1,8) vs. (2,4) vs. (4,2), etc. In summary, without a related work section that clearly contrasts this work with previous work, it is hard to judge the novelty of the proposed method. In addition, the experiment section could be improved in order to better understand the depth-vs-width tradeoff (since "width" is claimed to be an alternative to depth in the abstract).

Correctness: The method for density estimation appears to be correct, although in Eq. 5, the gradient should be wrt. G and not \phi? The result appears correct, though. The method for variational inference might be correct, but it is not entirely clear how the right-hand side of (10) is obtained? Was the reparameterization trick or REINFORCE somehow used? Is there an expectation missing in the result?

Clarity: Overall, the paper is nicely written. See Correctness section for some issues on clarity of the method.

Relation to Prior Work: See Weaknesses section.

Reproducibility: Yes

Additional Feedback: If your method is truly novel (other than using normalizing flows as the base learner), I would clarify how in the paper: Add a related work section which contrasts your method with Boosting Density Estimation [52], Boosted Generative Models [27], and boosting methods for variational inference, such as [6, 28, 41, 48]. To make room for this you could would move the section "5 Discussion" to appendix (and refer to it in the main text). Unfortunately, as it stands now, I think the paper is not ready for publication and thus lean towards rejection. UPDATE: Thanks for your responses. I continue to believe that the paper is incremental in the sense that few new ideas are introduced, although you do discuss some points that are specific to boosting flows. Furthermore, I think the paper will improve from a clearer discussion of related work (as promised in rebuttal). As a consequence, I've raised my score to 5: I would still lean towards rejection, but I'm also fine with acceptance.


Review 2

Summary and Contributions: This paper proposes a boosting procedure for normalizing flows (NFs). The paper recognizes that a convex combination of flows is itself a bijection and thus can be treated as simply another ‘step’ in the flow. Components (traditional NF architectures) are combined using the mixture formulation in Eq 3. Optimizing the new component and the mixing weight is done via the Frank-Wolfe algorithm. First the component is fit and then the mixing weight. Experiments are reported for density estimation and variational inference.

Strengths: As far as I know, dynamically adjusting the capacity of flows has not been previously explored for discrete-step architectures (the ODE solver can perform this function for continuous-time flows). Since coupling-layer architectures can be quite inefficient w.r.t. depth (e.g. Glow uses 100+ steps), the proposed procedure does offer a nice way of performing parallel evaluation that would allow for easier multi-GPU implementation.

Weaknesses: No increase in theoretical flow expressivity: Unlike traditional boosting in which an ensemble of weak learners is provably more expressive, the paper doesn’t provide such a proof for the proposed NF boosting procedure. Moreover, I conjecture that this methodology (in the general case, under NN / polynomial universal approximation assumptions) *cannot* build an ensemble that is more expressive than a single constituent component. There are two bottlenecks in NF expressivity---the base distribution and the class of transformation function [Papamakarios et al., 2019]---and the proposed method does not fundamentally change either of these. For example, the base distribution is simple and shared across all components (line 99). Recent work that does improve flow expressivity must use mixture formulations [Papamakarios et al., 2019] (discrete [Dinh et al., 2019] or continuous [Cornish et al., 2020] indices) whose base distribution (or support) and transformation change according to the index. Moreover, I suspect the proposed method shows some empirical improvements primarily because only RNVP and Glow models are used as the base components. These are the least efficient flows wrt depth, needing O(D) steps to achieve universal expressivity [Papamakarios et al., 2019]. On the other hand, a single step of neural autoregressive or sum-of-squares polynomial flow is a universal approximator. Yet these flows were not used in the experiments, and I expect the proposed technique would be much less effective for them. Of course, there is still worth in a boosting algorithm that can increase flow performance in practice. It’s just that the above issues are quite fundamental and not discussed by the paper at all or factored into the experimental design. Runtime analysis and demonstration of distributed evaluation: Given the theoretical issues above, I see the paper’s primary contribution as presenting an algorithm that allows for easy parallel evaluation of flows. Glow often requires a multi-GPU implementation to achieve state-of-the-art performance. This is problematic for high-performance settings in which we need to generate samples as quickly as possible. The proposed method could offer sampling time improvements because all GPUs could be run in parallel, one for each component. However, the paper does not perform any experiments isolating the runtime performance of the proposed method. Cornish, Rob, et al. "Relaxing bijectivity constraints with continuously indexed normalising flows." ICML (2020). Dinh, Laurent, et al. "A RAD approach to deep mixture models." arXiv preprint arXiv:1903.07714 (2019). Papamakarios, George, et al. "Normalizing flows for probabilistic modeling and inference." arXiv preprint arXiv:1912.02762 (2019).

Correctness: The paper claims that it proposes "a technique for increasing the flexibility of flow-based models" (line 249). This is incorrect in the general sense, and the paper needs to be much more precise in its claim. The contribution is practical. The proposed technique *does not* expand the theoretical class of target distributions that the flow can represent.

Clarity: No, the writing could be improved. The core methodology is presented quite summarily, in about one page.

Relation to Prior Work: No, this paper does not discuss its relationship to mixtures of flows, e.g. [Dinh et al., 2019], [Cornish et al., 2020].

Reproducibility: Yes

Additional Feedback: POST REBUTTAL UPDATE: Thank you, authors, for your responses. Indeed, I misunderstood, thinking you were combining the transformation functions, not the density functions. The issue of theoretical expressivity is not as severe as I previously thought since the method would indeed boost a 'weak' base NF. I think the paper should add some discussion of the expressivity classes of flows (see for ref [Jaini et al., ICML 2019]) and work on mixtures of flows [Dinh et al., 2019; Cornish et al., 2020]. But otherwise, I think the paper is ready for acceptance and have raised my score accordingly.


Review 3

Summary and Contributions: The paper applies the classic boosting approach to normalizing flows (NFs), whereby a mixture of NFs is trained by iteratively adding new components.

Strengths: The boosting approach seems to be a completely new idea in the field of NFs. The idea is easy to grasp and well motivated, but not trivial to come up with or put into practice (see e.g. 5.2 for some unique problems that have to be considered). For the experiments shown, including direct density estimation on toy data, low-dimensional unstructured data, as well as NFs in the variational inference setting (IAF, etc.), it works extremely well, considering the elegance and simplicity of the resulting model. I do wonder if it is the right way forward for NFs, particularly for competitive image generation, or if other approaches may prove more fruitful in the future. But this should not be for the reviewer to speculate, and does not affect my score. From a scientific standpoint, the work has excellent quality. The derivation is clear, caveats and limitations are addressed. Working code with decent quality is provided. Experiments are clearly explained, sensibly executed, and errors reported for all results.

Weaknesses: In my eyes, the density estimation part of the method seems like a fairly straight forward extension of [27], but using NFs instead of sum-product-nets or GMMs in data space. This makes me hesitant to give an even higher score.

Correctness: To my best judgement, the derivations and experiments are all correct.

Clarity: The writing is mostly easy to understand, the mathematical notation is sensible and precise. At a few points, I found the word choices or sentence structure strange, but not to the extent that it is a problem. Perhaps it could be proof-read again for this aspect in particular before the camera-ready version.

Relation to Prior Work: There is no dedicated 'Related Work' section, but the most relevant existing works are addressed in Line 75-81.

Reproducibility: Yes

Additional Feedback: ================== Update after Rebuttal ================== In their reviews and the discussion, the other reviewers raised some valid criticisms. Therefore, I will lower my score. I still find this a well thought out and scientifically sound paper, and think it should appear at NeurIPS.


Review 4

Summary and Contributions: Combine gradient boosting and normalizing flows for (1) density estimation and (2) variational inference. Gradient Boosting of Normalizing Flows allow constructing more expressive models by constructing a "wider" model instead of constructing a "deeper" model. The resulting mixture model can be evaluated in parallel, a big advantage compared to the "deeper" models.

Strengths: Most previous work on normalizing flows attempt to construct deeper models. This article presents an alternative. Instead of constructing deeper models the article presents a way of constructing expressive shallow models using boosting.

Weaknesses: Theoretical Grounding: No comments, the theoretical grounding seemed solid to me. Empirical Evaluation: The empirical evaluation was very thorough, however, the largest data set was 28x28 like MNIST. I imagine scaling to CIFAR with variational inference was hard. Technically, I guess this is a current limitation of GBNF. That said, based on the other 9 experiments, my guess is GBNF would scale to CIFAR, so this did not affect my score. Significance and novelty: No comments, the work seems significant and novel, see 'Strengths'.

Correctness: Theoretical claims: I believe the main equations (4), (7) and (12) are correct. That said, it possible I missed something. Empirical methodology: The empirical evaluation is quite thorough. GBNF is run on 9 datasets with different data modalities and the baselines seem fair. Tables also report \mu+-\sigma over 3 runs.

Clarity: I had a very easy time reading the paper. I wrote some comments on my first pass during the paper. I think you might find them useful, so I attached them below. L19-21: "Beyond their wide-ranging applications, generative models are an attractive class of models that place strong assumptions on the data and hence exhibit higher asymptotic bias when the model is incorrect [1]. " This sentence confused me a bit. I am not sure what it is trying to say, and why it is important to say this in the introduction. L18-24. Overall I like it, no comments. L 25. "Normalizing flows (NF) are an important recent development, " It sounds like NF were a recent development relative to the VAE, because the paragraph above talked about VAEs. But the VAE [37] was at ICLR December 2014 and NF [55,56] was published 2013 and 2010 respectively (using citation number as in article). The popularity of NF is more recent than that of VAE, but VAE are actually "more recent". L25-33: Overall I like it, no comments. L35-36: "With greater model complexity comes a greater risk of overfitting while slowing down training, prediction, and sampling. " A minor comment wrt. the risk of overfitting for larger models. I read a blog post that suggests training can be speed up by increasing model size [1]. L 37-39. Very short and concise, very good. L40-44: Short, very concise. The goal is clear. L45. "However, unlike a mixture model, GBNF offers the *optimality* advantages associated with boosting [2], " It is not clear what *optimality* means. I read optimal as in the sense of attaining least possible loss, a guarantee I don't think boosting has. Maybe it would be better to remove *optimality*? That said, it might be that there is just something I'm not understanding. L45-48: Looks good. L49: "Prediction and sampling not slowed with GBNF, as each component is independent and operates in parallel. " I do agree that parallelization improves, but I imagine the asymptotic time complexity increases linearly in the number of components, as is the case for Gradient Boosting. It sounds like it does not slow down in any way, which I think is wrong. L50-56: Ok. Background, L 58-82: I skimmed this section. The structure seems nice, and it seems to contains all the relevant background material. Figure 1: I think this is a really good figure that exemplifies GBNF. But you never refer to it, and it is usually good practice to refer to all figures. Suggestion: maybe refer to it in introduction, and move figure so it sits on page 2? Equation (3): It might've been easier if you started with the simple \psi(a)=a and \Gamma_K=1 and then later generalize. L106. I was a bit confused by the Franke Wolfe linear approximation part, maybe you could elaborate a bit on that.

Relation to Prior Work: There is no related work section, however, this is not an issue since the introduction adequately relates GBNF to previous work.

Reproducibility: Yes

Additional Feedback: Suggestion for future work. The AdaBoost algorithm satisfies a "training theorem". Informally, the theorem states that AdaBoost can perfectly classify 'n' points with O(lg n) weak learners under "mild" assumptions on the weak learners. Do you think "similar" theorems could hold for density estimators constructed using boosting? ** AFTER REBUTTAL** I went a bit back and forth during the reviewer discussion, I was sympathetic to some of the issues raised by the other reviewers, but finally, decided to keep my score at 7 accept.

[Author Response · NeurIPS 2020]

We want to thank the reviewers greatly for the time and effort put into these reviews. Each of you has provided a very detailed critique, and we appreciate your help in presenting our work as best as possible.

We are encouraged that reviewers found Gradient Boosted Normalizing Flows (GBNF) to be novel and significant (R3, R4), and that the "conceptual difference to previous research is a big strength." (R4). Reviewers appreciated that our "wider" approach to building normalizing flow-based models is more than just a way to improve performance, noting that sampling is trivial to parallelize across boosting components (R2, R4). Reviewers (R1, R3, R4) found the writing and derivations to be overall clear and concise (R4 kindly provides a few notes to improve clarity), and we are pleased that R3 found the proposed method easy to grasp but recognized GBNF as non-trivial to formulate or implement.

**[R1 ] "Contrast your method with Boosting Density Estimation, Boosted Generative Models, and boosting methods for variational inference"**  Our work uncovers challenges that are unique to boosting on normalizing flows. Specifically, for the variational inference setting we address three challenges in augmenting the VAE with a GBNF approximate posterior:

1. Only analytically invertible flows can be boosted for variational inference (Section 5.1, and Figure 2)
2. The "decoder shock" phenomena occurring from sudden changes to the VAE's posterior as new components are added (Section 5.2), and mitigating strategies.
3. Accelerating training by stochastically approximating the GBNF variational posterior (lines 148-152).

In regards to R1 and R3's critique on further differentiating our work with boosted density estimation [Rosset-Segal, '02] and generative models [Grover-Ermon, '18]: We show that the change-of-variables formula can be recursively computed in GBNF, and GBNF's mixture formulation remains invertible. Further, our work clarifies the narrative by providing a derivation for the boosting component updates and establishes a connection with Frank-Wolfe, whereas [Grover-Ermon, '18] merely state their choice of surrogate loss.

**[R2 ] "No increase in theoretical flow expressivity"**  We felt that proofs of boosting's expressiveness to be outside the scope of our paper. [Guo, '16] address the limit as the number of components $c \to \infty$, and [Cranko-Nock, '19] analyze properties of boosting for density estimation. While more work is needed in understanding these algorithms (R4 also suggests an important direction for future work), we focus on the *unique algorithmic challenges* of boosting normalizing flows for density estimation and variational inference.

We do, however, believe the composite model learned by GBNF is a fundamentally different class of transformation relative to the single component model. R2 writes "there are two bottlenecks in NF expressivity—the base distribution and the class of transformation function [Papamakarios et al., '19]—and the proposed method does not fundamentally change either of these." R2 references recent work [Dinh et al., '19, Cornish et al., '20] using mixture formulations (thank you, we will include these in our related works) that address the NF bottlenecks. GBNF also takes the form of a mixture, and offers a number of advantages over a standard mixture formulation:

1. GBNF is easier to train—we only need updates to the newest component (lines 102–109).
2. Easy to add more components (and flexibility) to an existing model, costing only additional training time (lines 95–97).
3. GBNF can learn a variational posterior, unlike [Cornish et al., '20].
4. GBNF optimizes a different objective that fits new components to the residuals of previously trained components (Eq. (6), lines 143-147), which can refine the *mode covering* behavior of maximum likelihood (similar to [Dinh et al., '19]) and VAEs (Figure 1).

**[R1 ] Correctness of (10)**  We appreciate reviewers taking the time to check for correctness. We stand by Eq. (10), and clarify that we are taking the gradient of the loss (9) with respect to $G^{(c)}$ at the point $\mathbf{z}$ in the function space, as opposed to the full distribution over random variable $\mathbf{z}$. We are left with the fixed components $G_K^{(c-1)}$ in the denominator because for a small step-size $\rho_c$ we have $G^{(c)}|_{\rho_c \to 0} = (1 - \rho_c)G^{(c-1)} + \rho_c g^{(c)}|_{\rho_c \to 0} = G^{(c-1)}$.

**[R1, R3, R4 ] A dedicated Related Work section**  We agree that a separate Related Work section will improve the readability of our manuscript. At present, comparisons to related work are scattered: relevant boosting approaches are listed in the Background section (lines 75–81), and in Section 5.1 many state-of-the-art flows are discussed and assessed based on compatibility with GBNF. We will reformat our manuscript to highlight these related works, and clearly differentiate our work from boosted density estimators [Rosset-Segal, '02, Grover-Ermon, '18], and normalizing flows using mixture formulations [Dinh et al., '19, Cornish et al., '20].

[Meta-Review · NeurIPS 2020]

The paper describes a way to create mixtures of normalizing-flow models using gradient boosting. Combining several simple flow models is an alternative to increasing the capacity of a single model, and is worth exploring. One of the main concerns the reviewers expressed is that of limited novelty, in that the proposed method is largely an application and continuation of existing techniques. However, the reviewers agree that the paper is well written, well executed, that although the idea is incremental there are still things to be said about applying gradient boosting to flows, and that the experiments are well done. For these reason, I'm happy to recommend acceptance of the paper. I would strongly encourage the authors to take to heart the reviewers' feedback when preparing the camera-ready version, and improve upon related work as discussed in the rebuttal.